# Data assimilation cycle length and observation impact in mesoscale ocean forecasting

Paul Sandery[1]

[1]CSIRO Oceans and Atmosphere, Castray Esplanade Battery Point TAS 7008

**Correspondence:** Dr Paul A Sandery (paul.sandery@csiro.au)

**Abstract.** A brief examination of the relationship between data assimilation cycle length and observation impact in a practical global mesoscale ocean forecasting setting is provided. Behind real-time reanalyses and forecasts from two different cycle length systems are compared and skill is quantified using all observations typically available for ocean forecasting. A 1-day Ensemble Optimal Interpolation (EnOI) cycle is compared to a 3-day cycle. The mean analysis increments for the 1-day system are significantly smaller suggesting a less biased system. Comparison of mean absolute increments identifies observations have greater impact in the 1-day system. Whilst smaller mean increments and greater observation impact do not guarantee a better forecast system, analysis of 7-day parallel forecasts show that the 1-day cycle system delivers improvement in predictability when compared to all available independent observations. The results are dependent on region, model and observing system, however, show the 1-day cycle provides an overall improvement in predictability, particularly in the subsurface. This appears to mainly come from less biased initial conditions and suggests greater retention of memory from observations and improved balance in the model.

## 1 Background

Cycle length in sequential data assimilating forecasting systems is an important setting that relates to dynamical scales resolved by the numerical model and the observation system. Many ocean forecasting systems, for example those described in Cummings and Smedstad (2014); Martin et al. (2007); Chassignet et al. (2009); Ferry et al. (2010); Bertino et al. (2008), make different choices around cycle length. Shorter cycle-length implies more frequent analyses and initialisation of the dynamical model. This may not necessarily lead to a better forecast system. In multivariate systems, observed variables project onto unobserved variables and systems tend to perform best when model error covariances are adequately sampled and there is reasonable coverage of multiple observation types. Longer cycles favour better coverage, however, can introduce larger analysis increments, temporal representation errors and overfitting of observational data. Bias is a fundamental problem in atmospheric and ocean forecasting affecting system performance. Bias arises within an assimilation cycle shared by issues related to the assimilation system, the model and observations. Identifying the cause of bias can be almost impossible (Houtekamer and

Zhang, 2016). Mean analysis increments are sometimes used to detect model bias (Houtekamer and Mitchell, 2005; Oke et al., 2013b) and some bias correction schemes are based on this (Zhang et al., 2016; Takacs et al., 2016; Ha and Snyder, 2014). Some care must be taken when using mean analysis increments as a proxy for model bias as they are dependent on the structure of the background covariances and also contain observation bias (Dee, 2005). Furthermore they can be approximately zero and relatively meaningless in regions of few or no observations or when large errors of opposite sign cancel out over time. Provided observation coverage is sufficient, observation bias is minimal and background error covariances are physically meaningful, well sampled mean analysis increments can be a reasonable indication of model bias. Dee and Da Silva (1998) illustrated that mean analysis increments tend to underestimate forecast bias. This is because they depend on the rate and period of growth of perturbations, i.e. model error growth, so they are forecast lead-time and cycle length dependent. This questions the use of mean analysis increments to estimate and compare the bias of forecasting systems with different cycle lengths. It appears that the cycle length, however, should be based on that which is best for predictability. Aspects of this are touched on by running twin experiments with a global ocean forecasting system using cycle lengths of 1 and 3-days. The system used in this study is the current Bureau of Meteorology Ocean Model Analysis and Prediction System (OceanMAPS) version 3. Previous versions of this system are documented in Brassington (2013) and Brassington et al. (2007). OceanMAPS is global eddy resolving, forced by Numerical Weather Prediction (NWP), runs on a 3-day data assimilation cycle and carries out 7-day forecasts. It is able to constrain aspects of the mesoscale variability to the available real-time observations. It produces forecasts of synop-tic features of the ocean circulation, such as the locations of eddies and fronts, daily changes in sea surface temperature and mixed layer depth, wind driven surface flows and coastal trapped waves. As typical for ocean forecast systems like Ocean-MAPS, the largest errors tend to occur in regions of most rapidly growing dynamical instabilities (O'Kane et al., 2011), such as western boundary currents and along the Antarctic Circumpolar Current (ACC). Some of these features and the character-istic spatio-temporal scales resolved by the model are captured in Figure 1, which presents a snapshot of Sea Level Anomaly (SLA) for the 9th September 2013. The behind real-time forecasted SLA is shown with unassimilated forward independent super-observations for the same day from the 1-day cycle system. Information regarding the use of forward super-observations for forecast verification, as used in this study, can be found in Sakov and Sandery (2015) and Sandery and Sakov (2017).

## 2  Data and Methods

The Geophysical Fluid Dynamics Laboratory (GFDL) Modular Ocean Model version 4.1 (MOM4p1) (Griffies et al., 2009) is used. This is a Boussinesq three-dimensional primitive equation volume conserving ocean model. The OceanMAPS grid has 0.1° horizontal resolution and is the same as the Ocean Forecasting Australia Model version 3 (OFAM3) (Oke et al., 2013a), which is based on bathymetric data from Smith and Sandwell (1997). The grid has 51 vertical levels and the top cell approximates quantities at 2.5 m depth with the average resolution in the upper 200 m being approximately 10 m. The physical model settings include the use of a 4th-order Sweby advection method and a scale dependent isotropic Smagorinksy biharmonic horizontal mixing scheme as described in Griffies and Halberg (2000). The General Ocean Turbulence Model (GOTM) $\kappa$-$\epsilon$

scheme is used for vertical mixing. Note that tides are not explicitly modelled, rather a parameterisation of tidal mixing is implemented using the scheme of Lee et al. (2006).

Initial conditions for both systems are the same and taken from the multi-year OFAM3 spin-up for the 1st January 2012. The 1 and 3-day cycle systems are spun-up with data assimilation over a 1-year period to the 1st Janaury 2013. Hindcasts are

continued throughout 2013 and a series of 7-day forecasts, 3-days apart, with identical base dates as illustrated in Figure 2 are carried out from 3rd January 2013. The forecast experiments were done behind real-time, therefore observations in the 12-24 hours prior to forecast base time were available to both systems, whereas in practice they would not be available in this period in a real-time system. The model is forced by 3-hourly prescribed surface fluxes of momentum, heat and salt from the Bureau of Meteorology operational global NWP system version 1 which is known as ACCESS-G APS1 (Australian Community

Climate and Earth System Simulator). For data asimilation the EnKF-C software (Sakov, 2014) is used in Ensemble Optimal Interpolation (EnOI) (Evensen, 2003) mode. The analysis equation and background error covariances can be written as

$$\mathbf{x}^a = \mathbf{x}^f + \mathbf{B}\mathbf{H}^T \left[ \mathbf{H}\mathbf{B}\mathbf{H}^T + \mathbf{R} \right]^{-1} \left[ \mathbf{y} - \mathcal{H}(\overline{\mathbf{x}^f}) \right], \tag{1a}$$

$$\mathbf{B} \equiv \mathbf{A}\mathbf{A}^T \left[ (\mathbf{m} - \mathbf{1}) \right]^{-1}, \tag{1b}$$

where $\mathbf{x}^a$ and $\mathbf{x}^f$ are analysis and forecast state vectors respectively; $\mathbf{y}$ is an observation vector; $\mathbf{H}$ is a linear observation

operator, i.e. $\mathbf{H} = \nabla \mathcal{H}(\mathbf{x})$, where $\mathcal{H}$ is a linear affine observation operator; $\mathbf{B}$ is background error covariance; $\mathbf{R}$ is observation error covariance; $\mathbf{A}$ represents ensemble anomalies; $\mathbf{m}$ is ensemble size and $^T$ denotes matrix transposition. $\mathbf{x}^f$ is taken to be an instantaneous model state, whereas $\overline{\mathbf{x}^f}$ is a 1 day mean and 3 day mean in the respective systems. The system uses no nudging or incremental analysis updating (Ourmières et al., 2006), rather the model is directly initialised to the analysis. This approach allows the model to run the complete length of each cycle as a dynamical forecast without being influenced by forcing

from nudging terms in the model equations. It also includes any initialisation shock from imbalance in the analysis in order to assess the impact of this on forecasts. $\mathbf{B}$ is based on a 144 member ensemble of intra-seasonal (1-day minus bimonthly mean) anomalies generated from an 18 year run of OFAM3. A source of time filtering is implicit in the innovation vector $[\mathbf{y} - \mathcal{H}(\overline{\mathbf{x}^f})]$ from the fact that the super-observations tend to represent averages over the time window, particulary for observations with relatively larger coverage, such as SST. An asynchronous 3-day cycle FGAT (First Guess Appropriate Time) system was not

compared with the 1-day or 3-day cycle systems as FGAT did not provide any improvements over the synchronous 3-day cycle. Mean increments and forecast errors from FGAT were comparable to the 3-day synchronous cycle (not shown).

Both the 1-day and 3-day systems assimilate the same original observations only once. The following observations are converted to super-observations weighted by inverse error variance. Altimetric SLA is taken from from the Radar Altimeter Database System (RADS) (Schrama et al., 2000) using tide, mean dynamic topography and inverse barometer corrections.

SLA observations are limited to water depths greater than 200 m. Sea surface temperature (SST) retrievals from the NAVO-CEANO (May et al., 1998) and WindSat (Gaiser et al., 2004) databases are used. All available in-situ temperature and salinity observations on the Global Telecommunications System (GTS) are used. These include Argo profiles (Roemmich et al., 2009), Conductivity Temperature Depth (CTD) and eXpendable BathyThermograph (XBT) profiles. The EnOI systems are run in a cycle scheme that centres the observation window as shown in Figure 2. The amount of super-observations generated by

the system from the original observations for the 1-day system is larger than the 3-day system. The total number of super-observations used in 2013 is shown in Table 1. In the data assimilation, a 250 km localisation radius is used for all observation types. The mean sea-level from OFAM3 (Oke et al., 2013a) is used for the model's mean dynamic topography to assimilate along track SLA observations.

## 3  Results

Global forecast innovation errors for the 1 year behind real-time period for 2013 are provided in Table 1. These are based on forward unassimilated observations, which can be regarded as independent. The 1-day cycle benefits statistically from a shorter forecast lead-time. These errors suggest an improvement in performance in constraining SLA, SST and sub-surface temperature and salinity. In order to determine whether this result is only dependent on forecast lead-time, a series of 44 parallel 7-day forecasts using identical base dates from 3rd January 2013 are analysed. These forecasts are compared to unassimilated observations. The global 7-day mean forecast errors are shown in Table 2 and Table 3 repeats this for the Tasman Sea region. It's interesting to note that whilst mean absolute deviation (MAD) global forecast errors are marginally smaller in the 1-day system that mean forecast bias is more significantly reduced for SLA, SST and sub-surface temperature. Figure 3 shows the global MAD forecast error growth as a function of lead time. Note that in order to ensure genuine forecasts are made the model is propagated to the end of the respective observation window in both systems, which is the position of the star in Figure 2. Daily mean forecast fields are saved and these are compared to the observations. For day zero, statistics are included that represent the errors in the initial conditions and the observation window partially overlaps half of this day in both systems so the statistics for day zero cannot be regarded as independent. The results suggest the 1-day system is better overall as a forecast system with improvements in lead time of about 1 day in surface variables and up to 7-days in sub-surface variables. The errors for salinity are relatively high for both systems as no restoring to salinity is used, however, the relative improvement is apparent.

The mean analysis increments for SST and SLA are shown in Figure 4. Three key features emerge regarding this estimate of model bias in the mean increments. There is an equatorial eastern Pacific Ocean cold bias, a southern ocean high latitude warm bias and mesoscale warm and cold biases in the western boundary current and ACC regions. Without speculating on the source of these systematic model errors it is noted that the first two aforementioned bias features have been detected in the CSIRO Climate Analysis Forecast Ensemble (CAFE) System, which is a configuration of the GFDL coupled model version 2.1 (CM2.1) run under an ensemble Kalman filter data assimilation framework. Figure 4 shows that mean increments are about one third smaller in the 1-day than the 3-day system, which can be expected for approximate linear error growth. The spatial patterns are very similar with the main difference being amplitude. Another way to compare increments over a period of time is to calculate the Mean Absolute Increment (MAI) (Figure 5). This is done in the following way. In each 3-day period the 1-day increments are summed and then the absolute values calculated. The mean of the absolute values over the 1 year period are then calculated. MAI for the two systems is only directly comparable if the forecast error growth is linear. The difference in mean increments between the two systems suggests this, however, error growth in the two systems is largest on the first

day (as seen in Figure 3) and becomes mainly linear after this. Regardless, the differences in spatial distribution of MAI for SLA and SST, shown in Figure 6, indicate the 1-day system has generally larger MAI. It can bee seen there is a greater impact from the observing system. The 1-day system projects more information from observed variables into unobserved variables through the background error covariances due to the relatively smaller observation coverage per analysis. For instance, in-situ observations from the Tropical Atmosphere Ocean - Triangle Trans-Ocean Buoy Network (TAO-TRITON) moored array in the equatorial Pacific Ocean produce larger MAI on SLA and SST in the 1-day system. It is also evident that the 1-day system has larger MAI in the western boundary currents and ACC. Figure 6 shows, as expected, that SST projects more into SLA in the 1-day system. SST observations in the 1-day system appear to be having greater impact in the regions of fastest growing dynamical instabilities. Interestingly, the relatively smaller MAI for SST in the 1-day system in the Inter-Tropical Convergence Zone (ITCZ), in the tropical warm pool in the western Pacific Ocean and at high latitudes in the Southern Ocean indicate the observing system is having less impact in these areas in the 1-day system.

Data assimilation typically injects energy into a forecast model as the observed fronts can be sharper than what can be supported by the model. In each cycle we usually see a jump in total kinetic energy with subsequent diminishing until the end of the forecast. This can be caused by factors such as insufficient horizontal and vertical resolution and imbalances in the analysis. Figure 8 shows total kinetic energy at 6 hourly temporal resolution. Here it can be seen that data assimilation in the 1-day system renders the state at a higher kinetic energy level, with smaller amplitude temporal fluctuations between cycles. The latter reflects the smaller increments per cycle in the 1-day system, however, the larger kinetic energy state indicates that more energy is retained in the mesoscale eddies, which infers that the gradients in SLA are maintained closer to observations. The larger MAI for SLA and SST in the 1-day system in the western boundary current and ACC regions reflects that observations are having a larger impact in these regions. The total kinetic energy dissipation for both systems in 2013 was calculated by summing the dissipation within each cycle and removing the trend. The 1-day system total kinetic energy dissipation is $8.4 \times 10^{18}$ J and that for the 3 day system is $9.6 \times 10^{18}$ J. The relative total kinetic energy dissipation, estimated by subtracting the mean dissipation from the respective systems, shows that the 1-day system has approximately 17% less relative kinetic energy dissipation than the 3-day system suggesting it is more effective at preserving SLA gradients and may be more dynamically balanced.

## 4 Conclusions

Global errors from a set of 44 parallel 7-day forecasts over a 1 year period in 2013 showed the 1-day cycle system delivered improvements in predicting sea surface temperature, sea level anomaly, subsurface temperature and salinity. The difference in mean absolute increments between the two cycle length systems indicated that the same observations had a greater impact on the 1-day system, with a larger degree of observed variables projecting onto unobserved variables. Greater observation impact does not necessarily lead to an improved forecast system as overfitting observations can produce dynamical imbalances, which can have deleterious effects on forecasts. The results, however, indicate that the 1-day cycle takes greater advantage of the observations and, compared to the 3-day cycle, is less biased in initial conditions and forecasts. This suggests also that

the background error covariances are a reasonable estimate of model error. With the shorter cycle length data assimilation introduces a larger amount of kinetic energy from the observations into the state, bringing the model closer to a realistic representation of ocean's kinetic energy. The 1-day cycle introduced a larger amount of information from the observations into the model with more frequent smaller adjustments at finer scales. The overall improvement in predictability, particularly in the subsurface, suggests greater retention of memory from observations and improved balance in the model. It is noted that, whilst an overall improvement in global performance was detected, in some regions the 1-day scheme may not perform better than the 3-day system. The results are a practical example of the influence of cycle length in global mesoscale ocean forecasting with the current observation network. The 1-day cycle is closer to asynchronous data assimilation and appears to be an improvement over the First Guess Appropriate Time (FGAT) approach (Cummings, 2005; Lee, 2005; Atlas et al., 2011) as our FGAT experiments did not yield as significant an improvement.

*Acknowledgements.* This work was carried out within the Bluelink Project with financial support from the Australian Bureau of Meteorology, Commonwealth Scientific and Industrial Research Organisation and Royal Australian Navy. Numerical simulations were undertaken using the Raijin supercomputer at the National Computational Infrastructure.

## 5  Additional Information

### 5.1  Code availability

The ocean model is available at https://github.com/mom-ocean/MOM4p1 and the data assimilation code can be found at https://github.com/sakov/enkf-c. These codes are documented within. The OceanMAPS3 system and observation processing scripts are intellectual property of the Bureau of Meteorology.

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

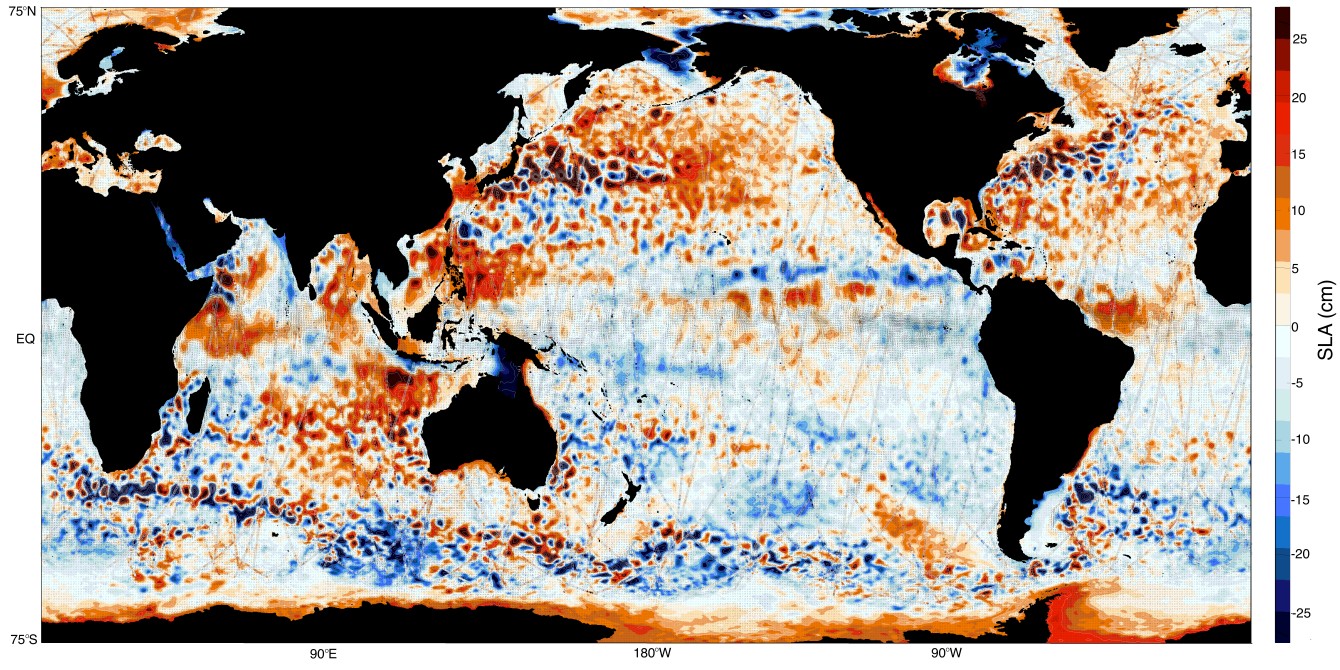

**Figure 1.** Forecast sea-level anomaly (SLA) from the 1-day cycle system for the 3rd of September 2013. Unassimilated forward independent super-observations are shown with coloured circles and grey outline on the same colorscale. The figure is high resolution and may be zoomed in for a detailed inspection of any region in the electronic version. Also shown are surface current vectors (black arrowheads) and surface wind vectors (blue arrowheads).

**Table 1.** Global mean behind real-time forecast innovation mean absolute deviation (MAD) and bias for sea-level anomaly (SLA), sea surface temperature (SST), sub-surface temperature (T) and salinity (S) from 1 year behind real-time period for 2013. See Figure 2 for cycle scheme. Total number of super-observations used in 2013 shown. † 1-day system ‡ 3-day system.

| Variable (units) | MAD$^{\dagger}$ | Bias$^{\dagger}$ | MAD$^{\ddagger}$ | Bias$^{\ddagger}$ | Observations$^{\dagger}$ | Observations$^{\ddagger}$ |
|---|---|---|---|---|---|---|
| SLA (cm) | 5.14 | 0.05 | 5.48 | 0.08 | 27070422 | 26033356 |
| SST (K) | 0.277 | 0.014 | 0.330 | 0.03 | 210063788 | 175258730 |
| T (K) | 0.517 | -0.0877 | 0.539 | -0.0934 | 6125208 | 5964116 |
| S (psu) | 0.13 | 0.0096 | 0.14 | 0.0104 | 5562515 | 5380711 |

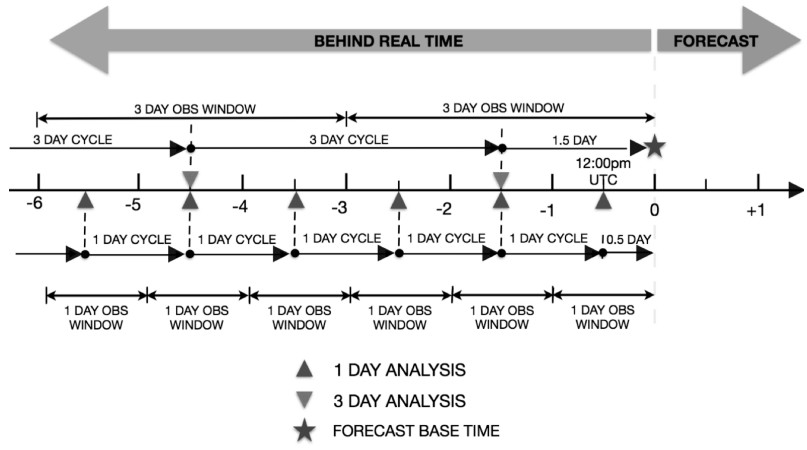

**Figure 2.** The analysis-forecast scheme used to compare the 3-day with the 1-day cycle system.

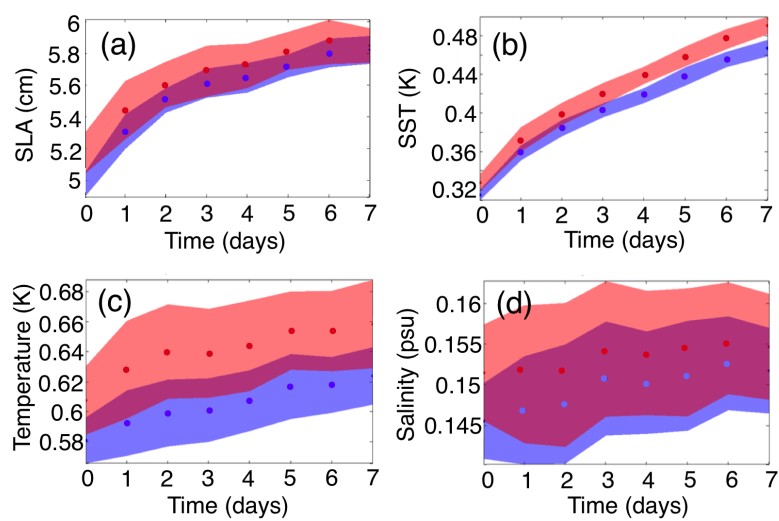

**Figure 3.** Global 7-day forecast innovation error statistics from series of identical base dates for (a) sea-level anomaly, (b) sea surface temperature, (c) sub-surface temperature and (d) sub-surface salinity. 1-day system shown in blue and 3-day system shown in red. The envelopes represent ± 1 standard deviation in forecast error. See Figure 2 for forecast scheme.

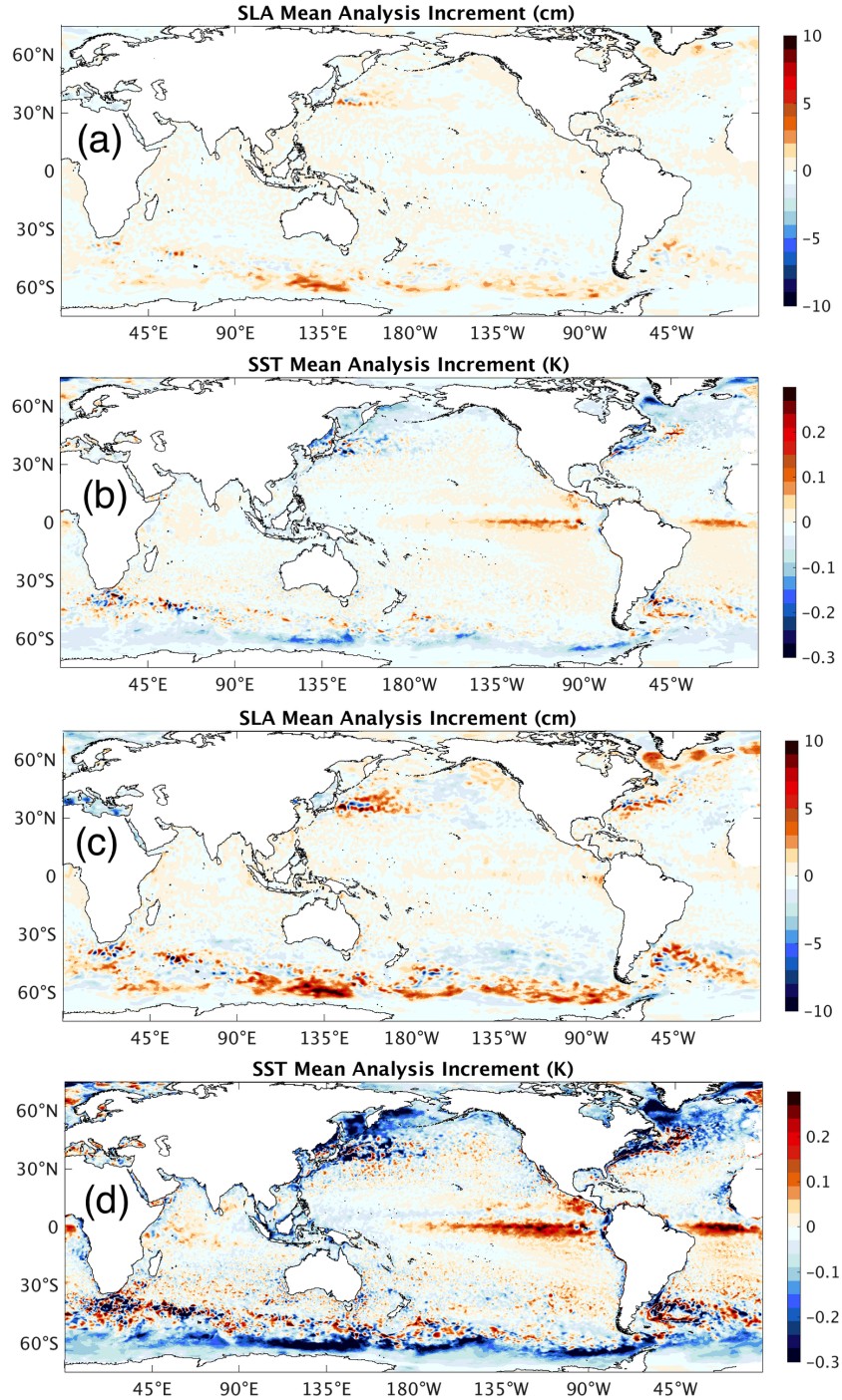

**Figure 4.** Mean analysis increments for sea-level anomaly (SLA) and sea surface temperature (SST) for the 1-day (a,b) and 3-day system (c,d).

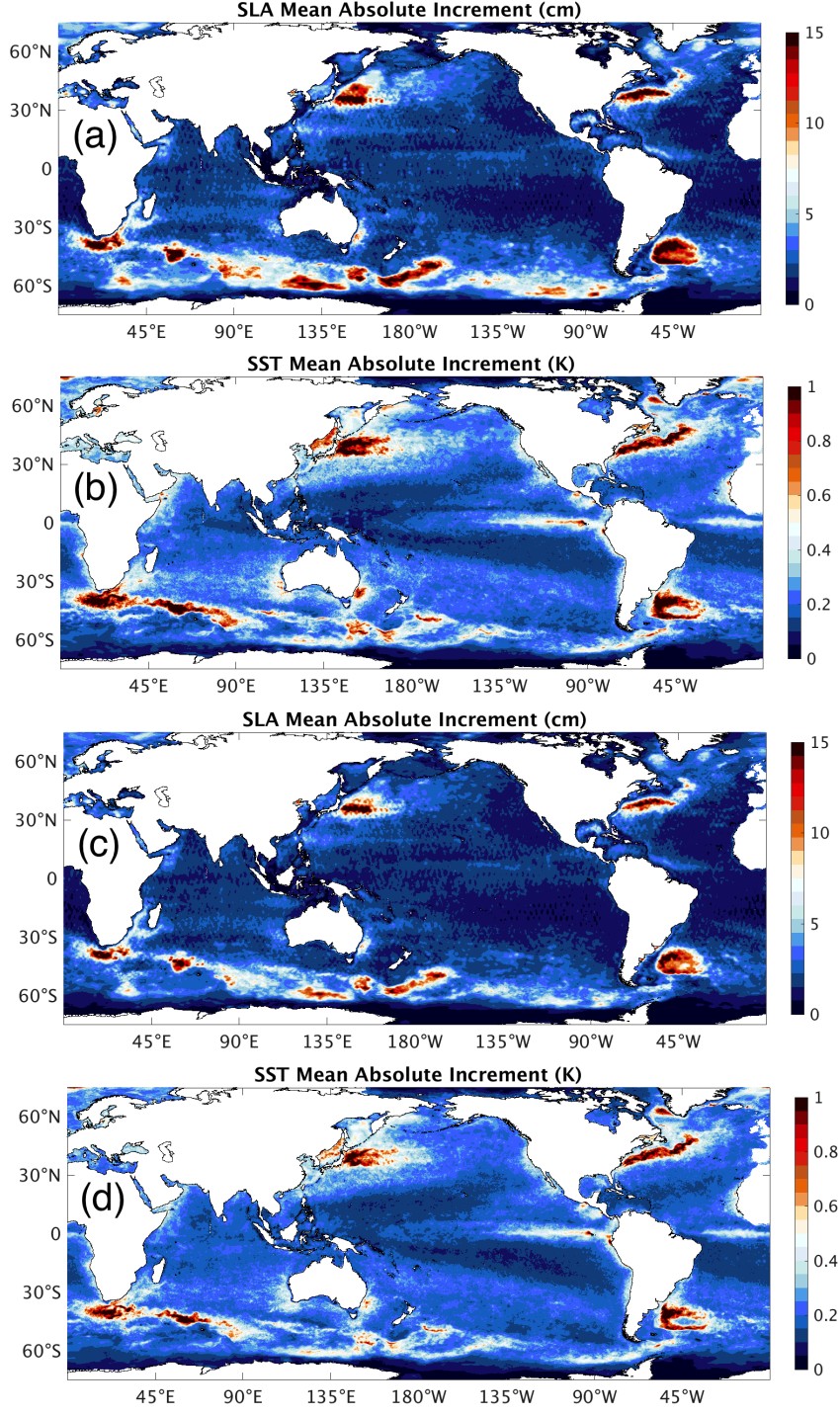

**Figure 5.** Mean absolute increments for sea-level anomaly (SLA) and sea surface temperature (SST) for the 1-day (a,b) and 3-day system (c,d).

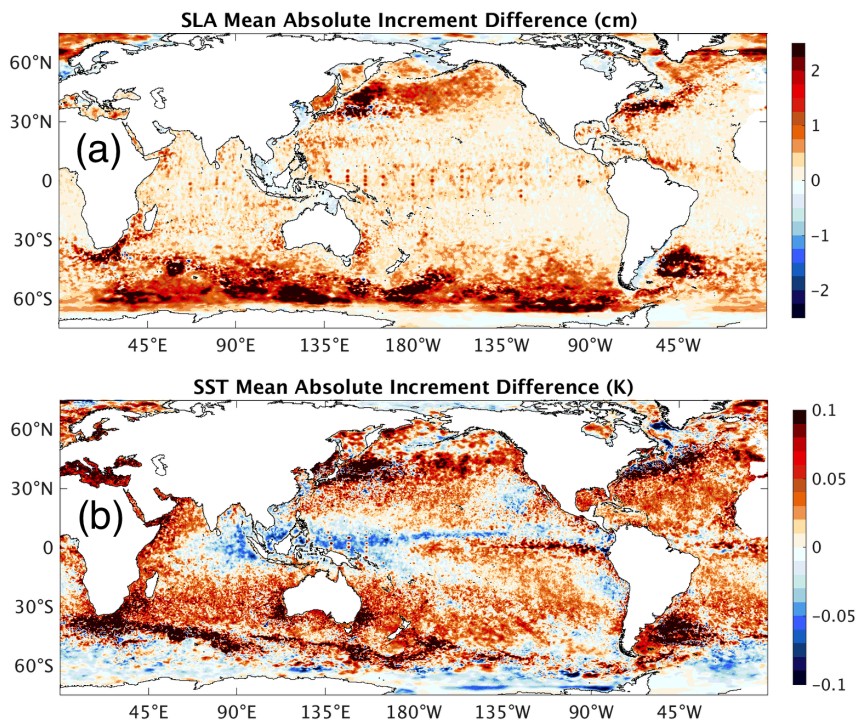

**Figure 6.** Difference (1-day minus 3-day) in mean absolute increment (a) for sea-level anomaly (SLA) and (b) sea surface temperature (SST).

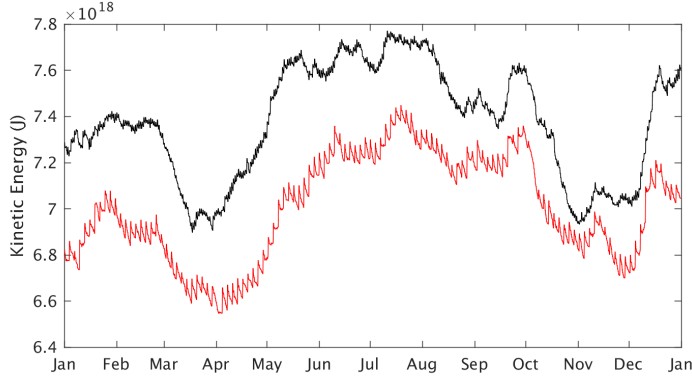

**Figure 7.** Total kinetic energy (Joules) for the 1-day (black) and 3-day systems (red) throughout 2013.

**Table 2.** Global mean and 7-day mean forecast innovation mean absolute deviation (MAD) and bias for sea-level anomaly (SLA), sea surface temperature (SST), sub-surface temperature (T) and salinity (S) from series of 44 7-day forecasts, 3-days apart from 3rd January 2013. See Figure 2 for information on how the base dates are aligned. ⋆ Total number of super-observations used to verify the 44 7-day forecasts shown. † 1-day system ‡ 3-day system.

| Variable (units) | MAD† | Bias† | MAD‡ | Bias‡ | Observations⋆ |
|---|---|---|---|---|---|
| SLA (cm) | 5.51 | 0.0152 | 5.60 | 0.197 | 21272458 |
| SST (K) | 0.417 | 0.0151 | 0.435 | 0.0457 | 237176982 |
| T (K) | 0.603 | -0.0979 | 0.616 | -0.136 | 5276357 |
| S (psu) | 0.153 | 0.0349 | 0.155 | 0.0341 | 4974538 |

**Table 3.** As for Table 2, except for Tasman Sea region. † 1-day system ‡ 3-day system.

| Variable (units) | MAD† | Bias† | MAD‡ | Bias‡ | Observations⋆ |
|---|---|---|---|---|---|
| SLA (cm) | 7.11 | 0.0674 | 7.21 | 0.0667 | 202548 |
| SST (K) | 0.478 | -0.0634 | 0.488 | -0.124 | 3145463 |
| T (K) | 0.573 | -0.067 | 0.617 | -0.119 | 48056 |
| S (psu) | 0.104 | 0.0674 | 0.098 | 0.0667 | 51144 |