# Peer review of "Data assimilation cycle length and observation impact in mesoscale ocean forecasting"

_Geoscientific Model Development, 2017_

## Referee Comment (RC1) · Anonymous Referee #1 · 3 May 2018

**General comments**

This paper investigates the impact of data assimilation window length on a short range ocean forecasting system. Current operational ocean forecasting systems use a range of different assimilation windows from 6 hours to 10 days, and yet there has been little work to look at the impact of different assimilation windows in one systems. This makes the topic of the paper novel and relevant for the ocean modelling community. As we move towards coupled ocean-atmosphere data assimilations the length of the time windows used in the ocean are likely to reduce to be consistent with atmosphere assimilation windows. This study may therefore be of particular interest to those developing coupled data assimilation systems. However, the results in the study would be more significant if the experiments were not using a synchronous data

assimilation method. The author should address this explicitly earlier in the paper. It's not clear that you couldn't just achieve similar improvements through asynchronous data assimilation.

The innovation statistics (Figure 6) from the 7 day forecasts are a significant result and strongly support the use of a 1 day window over a 3 day window in this system. However, the paper over emphasises the results from the mean and absolute mean increments. As alluded to in the background section, it is difficult to draw conclusions from comparisons of mean increments alone and the current organisation of the paper puts too much weight on the increment results. It would strengthen the interpretation of the assimilation increments if they were discussed within the context of the forecast statistics. I think that the paper could be substantially improved by presenting the innovation statistics first, as the key result, and providing the increments as supplementary evidence.

Throughout the paper the author states that the differences in mean absolute increments suggest observations are having a greater impact with a one day window. I think that you need to be careful with how this statement is used. A reader may mis-interpret this as meaning that larger increments automatically lead to an improved system. Presenting this result within the context of improved forecast innovations would make the statement more robust. The author should also clarify that larger increments do not necessarily mean a better data assimilation system.

In places the paper seems to lack details or justification. For example, the choice of forecast period for assessment or the choice of assimilation windows for the experiments. And is some places the paper seems to make contradictory conclusions about the results (particularly in relation to the mean increments). The paper should be modified to give a clearer narrative.

I think that there are some errors in the interpretation and description of the results. More details are given in the specific comments.

**Specific comments**

Abstract:

Page 1, line 4-5. The mean increments look to be approximately 1/3 smaller in the 1-day experiment, which is what you would expect for linear error growth. I don't think that you can make any statements about bias here without consideration of the error growth throughout the assimilation window. This statement is also inconsistent with your discussion of the mean increment results on page 4, line 8.

Page 1, line 9. I don't remember seeing any statistics which showed that the biggest improvements were in the Western Boundary currents.

Background:

Page 1, line 21-22. Over fitting is not just a problem for long data assimilation windows. In fact a long data assimilation window with good super obing or thining could produce smoother increments and be less influence by noise in the observations.

Page 2, lines 11. This paragraph is a bit confusing. It seems to argue that the mean increments are not a good indicator of bias, which contradicts your result on line 5 in the abstract. I didn't really understand what the purpose of this paragraph was. To justify the use of mean absolute increments?

Page 2, line 29. What is the forecast range of OFAM3? This might give more context to your choice of forecasts.

Page 2, line 30. Have you specified the horizontal resolution anywhere? This is important since the focus of the paper is mesoscale forecasting.

Page 3, line 10. "EnKF-C (Sakov, 2014) with Ensemble Optimal Interpolation (EnOI)" is not general terminology for a data assimilation scheme. This name is too

specific to be used without context. In reality, I think you are actually using EnOI?

Page 3, line 14. More details about the observation operator would be useful. You should also define the linear observation operator in equation 1.

Page 3, line 15. More details could be given on the data assimilation system, e.g clarifying that this is a synchronous data assimilation scheme, defining when in the time window the increments are applied (presumably the middle).

Page 3, line 23. You discuss the impact of super obing before introducing that you have used super obing. I think the order should be switched round.

Results:
Page 4, line 8. Seems contradictory to the abstract (page1, line 4-5)

Page 4, line 11. The MAI from the 2 experiments are only directly comparable if the forecast error growth is linear. It is worth discussing this here. Your results in Figure 7 should give some indication of the forecast error growth. From these figures it looks like the forecast error growth in the first day is a bit larger than subsequent days.

Page 4. It could also be useful to consider the variability in the increments.

Page 4, line 13-14. Would you expect the fact that you are assimilating more

observation in the 1-day experiment to also impact on the magnitude of the increments?

Page 4, line 16. What is the temporal resolution of the kinetic energy outputs in Figure 7.

Page 4, line 27. But also the model is only free running for 1 day before the next increments is applied, so less time to drift.

Page 4, line 28. Wouldn't the eddy kinetic energy be a better representation of the mesoscale energy?

Page 4, line 29. If you are going to claim that the model kinetic energy is closer to the observations, you should also show the observation kinetic energy. Comparing the results to the observations would also give more context for the difference between the two data-assimilation experiments. From the current Figure it's not clear how significant this increase in Kinetic Energy is.

Page 5, line 10. "mean forecast bias is more significantly reduced for SLA, SST and sub-surface temperature" - the mean forecast bias for SLA is actually slightly larger in the 1-day experiment in Table 3.

Page 5, line 13. Are the increments applied in the middle of the time window? Could you clarify this.

Page 5, line 10-11. Are the MAD statistics in Figure 6 calculated in the same way (using the same forecasts) as those presented in Table 2? Why do look so different? For example, the subsurface temperature MAD at day 7 looks to have a value of approximately 0.625, but in table 2 it's given as 0.603.

Conclusion:

Page 5, line 25-26. "Further 1 year runs of the two systems with an improved model using renanalysis bulk flux forcing have confirmed (not shown) that the 1-day cycle provides improvements in forecasting the mesoscale circulation in the western boundary current regions." Is this the evidence for the statement in the abstract that the biggest improvements are in the western boundary current region? You should show this result if it forms part of your main conclusions. It would, in general, be good to see more results focused on the mesoscale region given that the focus of the paper is mesoscale forecasting.

**Technical Corrections**

Page 9, Figure 1. There is quite a lot of irrelevant information on this plot which makes the key information difficult to see. The current vectors make the figure appear noisy in print, and they are not discussed anywhere in the paper. It would be best if they were removed.

Page 10, Figure 4. It would be better to use a sequential colour bar for Mean Absolute Increments.

Page 11, Figure 6 caption. typo throughout.
Page 11, Figure 7, (d). It's very difficult to see the results from the 3-day experiment for salinity.

---

## Referee Comment (RC2) · Anonymous Referee #2 · 21 May 2018

A brief examination of the relationship between data assimilation cycle length and observation impact in a practical global mesoscale ocean forecasting setting is provided.

This paper tries to address a practical issue but a very important one. The paper is well written and organized, and the results are useful for the broader modeling and forecasting community. Thus, the paper should be accepted for publication. However, there are multiple places that need to be corrected and/or changed, so a minor revision is justified.

Specific Comments: 1) Figure 1 is poorly displayed, at least in the paper version that will be used by most readers. "Unassimilated forward independent super-observations are shown with coloured circles and grey outline on the same colorscale": the grey outline is barely visible, while it is virtually impossible to see the coloured circles. Maybe

a second panel can be shown?

2) Figure 3 is confusing with three colors, blue, red and orange (?). I am guessing the red color is simply the overlap between blue and orange. This plot is relatively simple, and can be displayed by black-and-white lines (solid vs. dotted) showing a small bar or shaded lines representing the +/- 1 standard deviation.

3) p1, line 18, "...project into unobserved variables", replace "into" with "onto"

4) p2, line 5, replace "∼∼" with "approximately"

5) in the "Data and Methods" section, there is no place to mention the model grid size, which should be explicitly stated. I have to read the Oke et al., 2013a to find out this information as 1/10 degree.

6) p3, line 8, replace "3 hourly" with "3-hourly"

7) some acronyms not commonly used by the community are not necessary, e.g., MAI, MAD

8) Figure 2, only one star (representing forecast base time) is shown, should be displayed at the center of every 3-day cycle, right?

9) the font for some figures should be somewhat larger, particularly if multiple figures are printed on the same page

---

## Author Comment (AC1) · 19 Jul 2018

**18/06/2018**

I would like to take the opportunity to thank the reviewers for their constructive feedback, which helped improve the manuscript. Please find the response to their comments below. The following format has been adopted:

Reviewer comment

Author response

Text within the manuscript

**Anonymous Referee #1**

**General comments**

This paper investigates the impact of data assimilation window length on a short-range ocean forecasting system. Current operational ocean forecasting systems use a range of different assimilation windows from 6 hours to 10 days, and yet there has been little work to look at the impact of different assimilation windows in one systems. This makes the topic of the paper novel and relevant for the ocean modelling community. As we move towards coupled ocean-atmosphere data assimilations the length of the time windows used in the ocean are likely to reduce to be consistent with atmosphere assimilation windows. This study may therefore be of particular interest to those developing coupled data assimilation systems. However, the results in the study would be more significant if the experiments were not using a synchronous data assimilation method. The author should address this explicitly earlier in the paper. It's not clear that you couldn't just achieve similar improvements through asynchronous data assimilation.

The 1-day cycle is neither synchronous or asynchronous as we use daily binned observations. Over the years we have run experiments using asynchronous FGAT on a 3-day cycle and never found it to yield any improvements over synchronous DA. I have added a statement to the methods section regarding this.

An asynchronous 3-day cycle FGAT (First Guess Appropriate Time) system was not compared with the 1-day or 3-day cycle systems as FGAT did not provide any significant improvements over the synchronous 3-day cycle. Mean increments and forecast errors from FGAT were comparable to the 3-day cycle (not shown).

The innovation statistics (Figure 6) from the 7 day forecasts are a significant result and strongly support the use of a 1 day window over a 3-day window in this system. However, the paper over emphasises the results from the mean and absolute mean increments. As alluded to in the background section, it is difficult to draw conclusions from comparisons of mean increments alone and the current organisation of the paper puts too much weight on the increment results. It would strengthen the interpretation of the assimilation increments if they were discussed within the context of the forecast statistics. I think that the paper could be substantially improved by presenting the innovation statistics first, as the key result, and providing the increments as supplementary evidence. Throughout the paper the author states that the differences in mean absolute increments suggest observations are having a greater impact with a one day window. I think that you need to be careful with how this statement is used. A reader may misinterpret this as meaning that larger increments automatically lead to an improved system. Presenting this result within the context of improved forecast innovations would make the statement more robust. The author should also clarify that larger increments do not necessarily mean a better data assimilation system.

The forecast statistics are now presented first in the results section. There were statements in the abstract and manuscript that did explicitly say that greater observation impact and smaller mean increments do not imply a better system. Nonetheless, the abstract has been changed to

A brief examination of the relationship between data assimilation cycle length and observation impact in a practical global mesoscale ocean forecasting setting is provided. Behind real-time reanalyses and forecasts from two different cycle length systems are compared and skill is quantified using all observations typically available for ocean forecasting. A 1-day Ensemble Optimal Interpolation (EnOI) cycle is compared to a 3-day cycle. The mean analysis increments for the 1-day system are significantly smaller suggesting a less biased system. Comparison of mean absolute increments identifies observations have greater impact in the 1-day system. Whilst smaller mean increments and greater observation impact do not guarantee a better forecast system, analysis of 7-day parallel forecasts show that the 1-day cycle system delivers improvement in predictability when compared to all available independent observations. The

results are dependent on region, model and observing system, however, show the 1-day cycle provides an overall improvement in predictability, particularly in the subsurface. This appears to mainly come from less biased initial conditions and suggests greater retention of memory from observations and improved balance in the model.

The conclusion went on to say that

"The 1-day cycle system was shown to provide less biased initial conditions and improved forecasts, suggesting the relatively smaller mean analysis increments were reliably sampled."

This has been removed and the conclusions have been rewritten

In places the paper seems to lack details or justification. For example, the choice of forecast period for assessment or the choice of assimilation windows for the experiments. And in some places the paper seems to make contradictory conclusions about the results (particularly in relation to the mean increments). The paper should be modified to give a clearer narrative.

Justification for assimilation windows is based on the 3-day window behind real-time cycle and the 7 day forecasts that the current operational system uses and that the 3-day window is about the maximum length window that would be appropriate for assimilation into mesoscale eddy resolving system.

The following text has been changed to address this

"OceanMAPS is global eddy resolving, forced by Numerical Weather Prediction (NWP) and runs on a 3-day data assimilation cycle."

 to

"OceanMAPS is global eddy resolving, forced by Numerical Weather Prediction (NWP), runs on a 3-day data assimilation cycle and carries out 7-day forecasts"

In order to address some contradictions the following text in the abstract has been changed from

"This is thought to come from less biased initial conditions, greater observation impact and improved consistency with respect to the timing of model and observations."

to

"This appears to mainly come from less biased initial conditions."

I think that there are some errors in the interpretation and description of the results. More details are given in the specific comments.

**Specific comments**

Abstract:
Page 1, line 4-5. The mean increments look to be approximately 1/3 smaller in the 1-day experiment, which is what you would expect for linear error growth. I don't think that you can make any statements about bias here without consideration of the error growth throughout the assimilation window. This statement is also inconsistent with your discussion of the mean increment results on page 4, line 8. Page 1, line 9. I don't remember seeing any statistics which showed that the biggest improvements were in the Western Boundary currents.

The following statement regarding bias is in the context of error growth and is not inconsistent with other statements.

"… the mean increments are much smaller in the 1-day than the 3-day system. This is a natural result of shorter cycle length as model error growth is more constrained with more frequent analyses…"

This has been changed to

".. mean increments are about one third smaller in the 1-day than the 3-day system, which can be expected for approximate linear error growth."

Table 3 has robust statistics that show improvement in the bias in the Tasman Sea – where error is dominated by dynamical instabilities in the East Australian Current. Regardless, this is just one WBC region so I have removed the claims to better forecast skill for WBCs from the text.

Background:
Page 1, line 21-22. Over fitting is not just a problem for long data assimilation windows. In fact a long data assimilation window with good super obing or thining could produce smoother increments and be less influence by noise in the observations.

Point taken, however, in synchronous DA the longer the window, the more time averaging goes into the super-observation and mesoscale eddies become smeared out, less balanced, less accurate, less realistic.

Page 2, lines 11. This paragraph is a bit confusing. It seems to argue that the mean increments are not a good indicator of bias, which contradicts your result on line 5 in the abstract. I didn't really understand what the purpose of this paragraph was. To justify the use of mean absolute increments?

The paragraph sets the context for the problem, which is the reliability of the mean increment as a proxy for bias. Statements are made on the limitations then the following statement is made which sets the qualification.

"Provided observation coverage is sufficient, observation bias is minimal and background error covariances are physically meaningful, well sampled mean analysis increments can be a reasonable indication of model bias"

Page 2, line 29. What is the forecast range of OFAM3? This might give more context to your choice of forecasts.

OFAM3 was a 20-year spin-up run (free-model with atmospheric reanalysis forcing). Forecast range is not relevant to OFAM3.

Page 2, line 30. Have you specified the horizontal resolution anywhere? This is important since the focus of the paper is mesoscale forecasting.

This has now been specified

Page 3, line 10. "EnKF-C (Sakov, 2014) with Ensemble Optimal Interpolation (EnOI)" is not general terminology for a data assimilation scheme. This name is too specific to be used without context. In reality, I think you are actually using EnOI?

The sentence has been changed for clarification

 "For data asimilation the EnKF-C software (Sakov, 2014) is used in Ensemble Optimal Interpolation (EnOI) (Evensen, 2003) mode"

Page 3, line 14. More details about the observation operator would be useful. You should also define the linear observation operator in equation 1.

This has been addressed

Page 3, line 15. More details could be given on the data assimilation system, e.g clarifying that this is a synchronous data assimilation scheme, defining when in the time window the increments are applied (presumably the middle).

It has been clarified that the 3-day cycle is synchronous and the scheme is clearly shown in Figure 2.

Page 3, line 23. You discuss the impact of super obing before introducing that you have used super obing. I think the order should be switched round.

This has been rearranged and improved

Results:
Page 4, line 8. Seems contradictory to the abstract (page1, line 4-5)

This has been clarified

Page 4, line 11. The MAI from the 2 experiments are only directly comparable if the forecast error growth is linear. It is worth discussing this here. Your results in Figure 7 should give some indication of the forecast error growth. From these figures it looks like the forecast error growth in the first day is a bit larger than subsequent days.

Have added the following to address this

"MAI between the two systems is only directly comparable if the forecast error growth is linear. Forecast error growth in the two systems is largest on the first day (shown later in Figure 7)."

Page 4. It could also be useful to consider the variability in the increments.

The temporal RMS of the increments were initially looked at, and these were ~1/3 smaller in one day system similar to the mean increments. This was not considered to add anything and prompted looking for another metric (MAI).

Page 4, line 13-14. Would you expect the fact that you are assimilating more observation in the 1-day experiment to also impact on the magnitude of the increments?

Not necessarily as with the shorter cycle there is less error growth and generally smaller increments.

Page 4, line 16. What is the temporal resolution of the kinetic energy outputs in Figure 7.

This is 6 hourly. I have added this to the text

Page 4, line 27. But also the model is only free running for 1 day before the next increments is applied, so less time to drift.

OK

Page 4, line 28. Wouldn't the eddy kinetic energy be a better representation of the mesoscale energy?

Yes, the total kinetic energy in the systems represents this well as most of the kinetic energy in the ocean, and in the eddy-resolving ocean model used here, is in the mesoscale.

Page 4, line 29. If you are going to claim that the model kinetic energy is closer to the observations, you should also show the observation kinetic energy. Comparing the results to the observations would also give more context for the difference between the two data-assimilation experiments. From the current Figure it's not clear how significant this increase in Kinetic Energy is.

There are no observations of kinetic energy that would represent the kinetic energy in the model. One might derive KE from a geostrophic current product based on altimeter tracks (GSLA) but this has issues and is not representative of the model total KE. For example, GSLA does not project subsurface accurately, is unreliable on the shelf and can have over fitting of sparse altimeter tracks. The data assimilation system tells us that the analysis, which is closer to the observations, and satisfies more than just altimetry, has greater total kinetic energy every cycle, regardless of cycle length. The significance of the increase is important. We have not studied this in detail. The point been made here is that the signal is there and it makes a difference. Ultimately the improved forecast errors signify the 1-day system to be closer to observations.

Page 5, line 10. "mean forecast bias is more significantly reduced for SLA, SST and sub-surface temperature" - the mean forecast bias for SLA is actually slightly larger in the 1-day experiment in Table 3.

This statement is referring to global stats in Table 1. The mean forecast bias for SLA in Table 3 is for the Tasman Sea region, not really a problem. There is a sentence near the end that acknowledges this

"It is noted that, whilst an overall improvement in global performance was detected, in some regions the 1-day scheme may not perform better than the 3-day system"

Page 5, line 13. Are the increments applied in the middle of the time window? Could you clarify this.

The increments are applied at the analysis time. The observation window is centred as illustrated in Figure 2.

Page 5, line 10-11. Are the MAD statistics in Figure 6 calculated in the same way (using the same forecasts) as those presented in Table 2? Why do look so different? For example, the subsurface temperature MAD at day 7 looks to have a value of approximately 0.625, but in table 2 it's given as 0.603.

Table 2 provides the average of the MAD for the 7-day forecast. It should check out as the average of days 1-7 in Figure 6. The existing statement says we include day 0 (position of star in figure 2) in Figure 6 for illustration but do not include this in forecast stats as day 0 is not independent.

"For day zero, statistics are included that represent the errors in the initial conditions and the observation window partially overlaps half of this day in both systems so the statistics for day zero cannot be regarded as independent"

Conclusion:
Page 5, line 25-26. "Further 1 year runs of the two systems with an improved model using renanalysis bulk flux forcing have confirmed (not shown) that the 1-day cycle provides improvements in forecasting the mesoscale circulation in the western boundary current regions." Is this the evidence for the statement in the abstract that the biggest improvements are in the western boundary current region? You should show this result if it forms part of your main conclusions. It would, in general, be good to see more results focused on the mesoscale region given that the focus of the paper is mesoscale forecasting.

I have changed all references to improvements in WBCs to improvements in forecasting the mesoscale circulation.

**Technical Corrections**

Page 9, Figure 1. There is quite a lot of irrelevant information on this plot which makes the key information difficult to see. The current vectors make the figure appear noisy in print, and they are not discussed anywhere in the paper. It would be best if they were removed.

I would prefer that the original of Figure 1 is supplied to the reviewers and included in the final PDF. This unfortunately did not carry through to the reviewed manuscript. There is a wealth of information that can be clearly seen by zooming in.

Page 10, Figure 4. It would be better to use a sequential colour bar for Mean Absolute Increments.

Thanks for the suggestion. Figure 4 still shows what was intended.

Page 11, Figure 6 caption. typo throughout.

Fixed

Page 11, Figure 7, (d). It's very difficult to see the results from the 3-day experiment for salinity.

Figure 7, which is now Figure 3 has been improved.

**Anonymous Referee #2**

Specific Comments: 1) Figure 1 is poorly displayed, at least in the paper version that will be used by most readers. "Unassimilated forward independent super-observations are shown with coloured circles and grey outline on the same colorscale": the grey outline is barely visible, while it is virtually impossible to see the coloured circles. Maybe C1 a second panel can be shown?

The original submitted figure is high resolution and did not translate to as good resolution in the GMDD generated manuscript. My preference is the original be embedded in the final electronic version allowing readers to see the many interesting details within this figure. Most readers should be able to refer to the electronic version.

2) Figure 3 is confusing with three colors, blue, red and orange (?). I am guessing the red color is simply the overlap between blue and orange. This plot is relatively simple, and can be displayed by black-and-white lines (solid vs. dotted) showing a small bar or shaded lines representing the +/- 1 standard deviation.

Thanks for the suggestion The figure shows what was intended.

3) p1, line 18, "...project into unobserved variables", replace "into" with "onto"

Changed

4) p2, line 5, replace "~~" with "approximately"

Done

5) in the "Data and Methods" section, there is no place to mention the model grid size, which should be explicitly stated. I have to read the Oke et al., 2013a to find out this information as 1/10 degree.

Have explicitly now stated the horizontal resolution

6) p3, line 8, replace "3 hourly" with "3-hourly"

Done

7) some acronyms not commonly used by the community are not necessary, e.g., MAI, MAD

These have been defined once only at the first occurrence as appropriate for scientific writing.

8) Figure 2, only one star (representing forecast base time) is shown, should be dis- played at the center of every 3-day cycle, right?

This could be done, however, the message is more effectively communicated the way it is displayed as it shows that both cycles need to propagate the model past the last analysis to the end of the observation window.

9) the font for some figures should be somewhat larger, particularly if multiple figures are printed on the same page

---

## Author Comment (AC2) · 19 Jul 2018

The comment was uploaded in the form of a supplement:
https://www.geosci-model-dev-discuss.net/gmd-2017-298/gmd-2017-298-AC2-supplement.pdf

---

## Author Comment (AC3) · 31 Jul 2018

Abstract. A brief examination of the relationship between data assimilation cycle length and observation impact in a practical global mesoscale ocean forecasting setting is provided. Behind real-time reanalyses and forecasts from two different cycle length systems are compared and skill is quantified using all observations typically available for ocean forecasting. A 1-day Ensemble Optimal Interpolation (EnOI) cycle is compared to a 3-day cycle. The mean analysis increments for the 1-day system are significantly smaller suggesting a less biased system. Comparison of mean absolute increments identifies observations have greater impact in the 1-day system. Whilst smaller mean increments and greater observation impact do not guarantee a better forecast system, analysis of 7-day parallel forecasts show that the 1-day cycle

system delivers improvement in predictability, particularly for the subsurface. This improvement appears to mainly come from less biased initial conditions and suggests greater retention of memory from observations and improved balance in the model.

Please also note the supplement to this comment:
https://www.geosci-model-dev-discuss.net/gmd-2017-298/gmd-2017-298-AC3-supplement.pdf

**Supplement:**

[revised manuscript text omitted]